# Prevalence and Impact on Mortality of Colonization and Super-Infection by Carbapenem-Resistant Gram-Negative Organisms in COVID-19 Hospitalized Patients

**DOI:** 10.3390/v15091934

**Published:** 2023-09-15

**Authors:** Roberto Casale, Gabriele Bianco, Paulo Bastos, Sara Comini, Silvia Corcione, Matteo Boattini, Rossana Cavallo, Francesco Giuseppe De Rosa, Cristina Costa

**Affiliations:** 1Microbiology and Virology Unit, University Hospital Città della Salute e della Scienza di Torino, 10126 Turin, Italy; roberto.casale.cr@gmail.com (R.C.); cominisara@gmail.com (S.C.); matteo.boattini@unito.it (M.B.); rossana.cavallo@unito.it (R.C.); cristina.costa@unito.it (C.C.); 2Department of Public Health and Paediatrics, University of Torino, 10126 Turin, Italy; 3Independent Researcher, 1169-056 Lisbon, Portugal; pauloandrediasbastos01@gmail.com; 4Department of Medical Sciences, Infectious Diseases, University of Turin, 10126 Turin, Italy; silvia.corcione@unito.it (S.C.); francescogiuseppe.derosa@unito.it (F.G.D.R.); 5Lisbon Academic Medical Centre, 1169-056 Lisbon, Portugal; 6Unit of Infectious Diseases, Cardinal Massaia, 14100 Asti, Italy

**Keywords:** COVID-19, superinfection, rectal carriage, carbapenem resistance, *Acinetobacter baumannii*, KPC, ICU, mortality

## Abstract

Background: The relationship between superinfection by multidrug-resistant Gram-negative bacteria and mortality among SARS-CoV-2 hospitalized patients is still unclear. Carbapenem-resistant *Acinetobacter baumannii* and carbapenemase-producing Enterobacterales are among the most frequently isolated species when it comes to hospital-acquired superinfections among SARS-CoV-2 patients. Methods: Herein, a retrospective study was carried out using data from adult patients hospitalized for COVID-19. The interaction between in-hospital mortality and rectal carriage and superinfection by carbapenemase-producing Enterobacterales and/or carbapenem-resistant *Acinetobacter baumannii* was assessed. Results: The incidence of KPC-producing *Klebsiella pneumoniae* and/or carbapenem-resistant *Acinetobacter baumannii* rectal carriage was 30%. Bloodstream infection and/or pneumonia due to KPC-producing *Klebsiella pneumoniae* and/or carbapenem-resistant *Acinetobacter baumannii* occurred in 20% of patients. A higher Charlson comorbidity index (OR 1.41, 95% CI 1.24–1.59), being submitted to invasive mechanical ventilation/ECMO ≥ 96 h (OR 6.34, 95% CI 3.18–12.62), being treated with systemic corticosteroids (OR 4.67, 95% CI 2.43–9.05) and having lymphopenia at the time of admission (OR 0.54, 95% CI 0.40–0.72) were the features most strongly associated with in-hospital mortality. Conclusions: Although KPC-producing *Klebsiella pneumoniae* and/or carbapenem-resistant *Acinetobacter baumannii* rectal carriage, and/or bloodstream infection/pneumonia were diagnosed in a remarkable percentage of COVID-19 patients, their impact on in-hospital mortality was not significant. Further studies are needed to assess the burden of antimicrobial resistance as a legacy of COVID-19 in order to identify future prevention opportunities.

## 1. Introduction

Since the end of 2019, the severe acute respiratory syndrome coronavirus-2 (SARS-CoV-2) pandemic has represented a major burden to worldwide healthcare systems due to the high number of infected patients requiring lengthy hospitalization and intensive care [1]. The risk of COVID-19 mortality is higher in elderly patients, in males, in obese individuals, in those with underlying chronic organ diseases, in the presence of high inflammation or coagulation statuses, or in those with lymphopenia at admission [2]. In addition, SARS-CoV-2-mediated organ injury has been associated with prolonged mechanical ventilation, ICU, or hospital stays [3]. The former, together with antimicrobial misuse/overuse, may have paved the way to the dissemination of multidrug-resistant bacteria in these populations [4,5,6,7].

The intestinal tract represents a massive reservoir of potential pathogens that can, under certain circumstances, reach the bloodstream and/or extra-intestinal sites. Of particular note, the rise in multidrug-resistant bacterial intestinal carriage drives superinfections and mortality [8,9], which are particularly worrisome in vulnerable populations. Recently, a Spanish pilot study showed that the gut microbiota profile can become a predictive biomarker for multidrug-resistant bacteria colonization in SARS-CoV-2 patients [10]. Among Gram-negative bacteria, *Acinetobacter baumannii* and *Klebsiella pneumoniae* have been reported as the most frequently isolated species among COVID-19 patients with hospital-acquired superinfections [5,6]. The antibiotic resistance pattern of these pathogens is very diverse and dependent, among other factors, on local epidemiology and intensive care practices [7,11,12,13,14,15]. The data on carbapenem resistance for *Acinetobacter baumannii* have attracted particular attention given its independent association with 14-day mortality [11] and the risk of developing superinfections [14], and several outbreaks of ceftazidime/avibactam-resistant KPC-producing *Klebsiella pneumoniae* have also been reported [16,17,18,19].

The relationship between intestinal carriage or superinfection by multidrug-resistant Gram-negative bacteria and mortality among SARS-CoV-2 hospitalized patients is still unclear [7]. This study aimed to evaluate the impact of carbapenemase-producing Enterobacterales and carbapenem-resistant *Acinetobacter baumannii* rectal carriage and/or superinfection on COVID-19 in-hospital mortality.

## 2. Materials and Methods

### 2.1. Setting

This retrospective observational study was conducted at the “City of Science and Health of Turin”, a tertiary care teaching hospital with 1900 beds in Turin, North–West Italy. This region presents a high prevalence of Gram-negative bacteria with complex resistance phenotypes [19].

### 2.2. Study Design and Data Collection

This study included all patients (≥18 years old) admitted for laboratory-confirmed SARS-CoV-2 infection from March to June 2020. Variables herein analyzed included demographics, community acquisition of SARS-CoV-2 infection, comorbidities, intensive care unit (ICU) admission, hospital lengths of stay, lung infiltrates and/or pulmonary embolism on chest CT scan, blood tests at the time of admission, type of support and therapeutic management (antivirals, immunosuppressors, and antimicrobial treatment), carbapenemase-producing Enterobacterales and/or carbapenem-resistant *Acinetobacter baumannii* rectal carriage over the course of the admission, carbapenemase-producing Enterobacterales and/or carbapenem-resistant *Acinetobacter baumannii* bloodstream infection and/or pneumonia over the course of the admission, and in-hospital mortality. All data were extracted from electronic medical records.

### 2.3. Definitions

SARS-CoV-2 infection was defined as community-acquired if symptoms started within 72 h of admission. Chronic heart disease was considered only in the presence of class II NYHA disease or worse. Chronic pulmonary disease was defined as a disorder that affects the respiratory system, including asthma, chronic obstructive pulmonary disease or pulmonary fibrosis. Chronic kidney disease was considered only if KDIGO was stage 3A or worse. Chronic liver disease was defined as a progressive deterioration of liver functions for more than six months. The Charlson comorbidities index [20] was calculated at hospital admission. Bacterial superinfection was defined as a clinical deterioration with Systemic Inflammatory Response Syndrome signs and the concomitant presence of carbapenemase-producing Enterobacterales and/or carbapenem-resistant *Acinetobacter baumannii* identified in lower respiratory tract specimens or blood cultures after at least 48 h from the admission. Systemic corticosteroids included dexamethasone and other glucocorticoids.

### 2.4. Microbiological Diagnostics

The laboratory confirmation of SARS-CoV-2 infection was performed using AllplexTM 2019-nCoV assay (Seegene Inc., Seoul, Republic of Korea) on naso/oropharyngeal swabs or bronchoalveolar lavage fluid obtained from patients with signs or symptoms of SARS-CoV-2 infection. For patients with more than one positive test, the first episode was considered for study purposes.

The detection of rectal carriage of carbapenemase-producing Enterobacterales and carbapenem-resistant *Acinetobacter baumannii* for new admissions and inpatients was performed on a weekly basis. Rectal swabs were collected using the FecalSwab™ system (Copan, Brescia, Italy) and inoculated on a chromogenic screening plate (Chromatic CRE medium, Lioflchem, Roseto degli Abruzzi, Italy) by automated direct plating using the WASP^®^ instrument (Copan, Brescia, Italy). Species identification was carried out through MALDI-TOF MS analysis (Bruker Daltonics GmbH, Bremen, Germany), and carbapenemase production was investigated through genotypic testing (Xpert Carba-R assay; Cepheid Sunnyvale, CA, USA) and/or lateral flow immunoassay (NG-test CARBA 5; NG Biotech, Guipry, France). The laboratory confirmation of carbapenemase-producing Enterobacterales and/or carbapenem-resistant *Acinetobacter baumannii* bloodstream infection and/or pneumonia was based on: (1) MALDI-TOF MS identification of Enterobacterales and/or *Acinetobacter baumannii* on blood culture and/or lower respiratory tract specimens overnight subcultures, (2) detection of carbapenemase genes in Enterobacterales with Xpert Carba-R assay (Cepheid, Sunnyvale, CA, USA), and (3) detection of carbapenem resistance (meropenem and/or imipenem) in *Acinetobacter baumannii* isolates using the MicroscanWalkAway plus system (Beckman Coulter, Brea, CA, USA). Antimicrobial susceptibilities were interpreted according to the current EUCAST breakpoints [21].

### 2.5. Statistical Analysis

The Multivariate Imputation by Chained Equations algorithm was used to input the ~1.2% of missing laboratorial results for the six variables D-dimer, lactate dehydrogenase, creatine phosphokinase, NT-proBNP, troponin T, and ferritin. For each missing value, 50 chained models were averaged out to estimate the most likely values. Kernel density estimations were used to visualize the probability density function of both initial and imputed values and confirm that these had an almost perfect overlap. The imputation of missing values had an almost imperceptible effect on both the mean and median of the inputted variable. Summary descriptive statistics were presented as median [IQR] and proportion of positive cases. Normality was assessed using the Shapiro–Wilk test. In the case of categorical variables, proportions were compared using Fisher’s exact test. In the case of continuous variables, the Mann–Whitney U test was employed. Univariate and multivariate logistic regression models were fit to determine which features were significantly associated with in-hospital mortality. Variables with a *p*-value < 0.05 in univariate regression were further funneled into a multivariate logistic regression. Features pertaining to hospitalization time were deliberately left out of the model as these were not baseline predictors (i.e., could only be known after the fact) and were thus of questionable clinical relevance for future cases. Data analysis was carried out using R 4.2.2 software.

## 3. Results

### 3.1. Patients Characteristics

Overall, 188 patients were included in this study (Table 1). The median age was 69 years [IQR 67–75], 71% were male, and 65% suffered from a community-acquired SARS-CoV-2 infection. The main comorbidities observed were chronic heart disease (66%), diabetes (27%), chronic pulmonary disease (22%), obesity (20%), chronic kidney disease (19%), and active neoplasia (18%). The median Charlson comorbidity index was four [IQR 2–5]. In regards to the clinical presentation of a SARS-CoV-2 infection, 98% of the patients presented with bilateral pneumonia or acute respiratory distress syndrome (ARDS) criteria. During hospitalization, 86% of the patients underwent non-invasive ventilation and/or O2 therapy. Invasive mechanical ventilation was required for 50% of patients. Systemic corticosteroids and antimicrobial treatment were started in 81% and 96% of patients, respectively. For critically ill patients (61%), the median length of ICU stay was 10 days [IQR 5–18], while the overall median length of hospital stay was 29 days [IQR 18–46]. Thirty percent of patients presented with KPC-producing *Klebsiella pneumoniae* and/or carbapenem-resistant *Acinetobacter baumannii* rectal carriage during hospitalization, and 20% suffered from a bloodstream infection and/or pneumonia caused by these.

### 3.2. Comparison of SARS-CoV-2 Survivors vs. Non-Survivors Clinical Features

The patients who did not survive hospitalization were older (median age 73, IQR 65–77, *p*-value < 0.01), were more likely to suffer more from chronic respiratory disease (*p*-value = 0.02) and neoplasia (*p*-value = 0.03), presented a higher Charlson comorbidity index (*p*-value < 0.01), more often experienced bilateral pneumonia or ARDS (*p*-value = 0.01), and more often required intensive care (*p*-value = 0.01, Table 2). The blood counts of patients failing to survival hospitalization presented higher values of NT-proBNP (*p*-value = 0.01) and creatinine (*p*-value = 0.02) and lower lymphocytes counts (*p*-value < 0.01). In addition, these patients were more likely to undergo both invasive (*p*-value < 0.01) and non-invasive ventilation and/or O_2_ therapy (*p*-value = 0.02) and were more often administered systemic corticosteroids (*p*-value < 0.01). No statistically significant differences were observed for KPC-producing *Klebsiella pneumoniae* and/or carbapenem-resistant *Acinetobacter baumannii* rectal carriage and/or bloodstream infection and/or pneumonia between survivors and non-survivors in the univariate analysis.

### 3.3. Factors Associated with In-Hospital Mortality

The logistic regression analyses, when adjusting for all other possible confounding factors, showed the higher Charlson comorbidity index (OR 1.41, 95% CI 1.24–1.59), being submitted to invasive mechanical ventilation or ECMO ≥ 96 h (OR 6.34, 95% CI 3.18–12.62), being treated with systemic corticosteroids (OR 4.67, 95% CI 2.43–9.05), and having a lower lymphocyte count at the time of admission (OR 0.54, 95% CI 0.40–0.72) as the sole features significantly associated with in-hospital mortality (Table 3). KPC-producing *Klebsiella pneumoniae* and/or carbapenem-resistant *Acinetobacter baumannii* rectal carriage and/or bloodstream infection and/or pneumonia had no significant association with in-hospital mortality.

## 4. Discussion

Bacterial superinfections may result in significantly worst COVID-19 outcomes, including increased mortality [5,22]. Of particular note, carbapenem-resistant Gram-negative bacteria, such as carbapenemase-producing Enterobacterales and carbapenem-resistant *Acinetobacter baumannii*, represent a significant threat due to the limited availability of effective treatment options and the ease with which they spread across health care facilities [19,23,24]. This study aimed to compare in-hospital COVID-19 mortality in the presence/absence of carbapenemase-producing Enterobacterales and/or carbapenem-resistant *Acinetobacter baumannii* colonization and/or infection in an area with a heavy load of endemic multidrug-resistant pathogens.

KPC-producing *Klebsiella pneumoniae* and carbapenem-resistant *Acinetobacter baumannii* rectal carriage has been shown to increase the risk of subsequent infections [25], and we have herein observed that a considerable percentage (30%) of COVID-19 patients do present with KPC-producing *Klebsiella pneumoniae* and/or carbapenem-resistant *Acinetobacter baumannii* rectal carriage. KPC-producing *Klebsiella pneumoniae* was the most prevalent in the analyzed population, being identified in 27% of patients. For 35% of such cases, KPC-producing *Klebsiella pneumoniae* was present together with carbapenem-resistant *Acinetobacter baumannii*. Among the colonized patients, 20% developed subsequent pneumonia and/or a bloodstream infection. No carbapenemases other than KPC were identified among carbapenem-resistant Enterobacterales isolates. While during the early stages of the COVID-19 pandemic, unremarkable increases in the incidence of infections by carbapenemase-producing Enterobacterales were observed [26,27], such figures significantly changed later on [28]. Tiri et al. reported that the incidence of KPC-producing *Klebsiella pneumoniae* in COVID-19 ICU patients increased significantly from 6.7% in 2019 to 50% in 2020 [29]. Likewise, several carbapenemase-producing Enterobacterales outbreaks (including KPC-producing *Klebsiella pneumoniae*, ceftazidime/avibactam-resistant KPC-producing *Klebsiella pneumoniae* and NDM-producers) have been reported worldwide [16,17,18,30,31]. A review of studies on carbapenem-resistant *Klebsiella pneumoniae* in COVID-19 patients from six countries (Italy, China, Egypt, United States, Spain, and Peru) showed that 84% of infected patients were male, with a mean age of 61 years, and the predominant carbapenemases were KPC and NDM. Several factors contributed to the variable prevalence of carbapenemase producers, ranging from 0.35 to 53%, with the lowest prevalence reported in the United States and the highest in China [32]. In the present study, an incidence of 14% for KPC-producing *Klebsiella pneumoniae* superinfection was observed, with 6% of patients developing pneumonia and/or a bloodstream infection due to both KPC-producing *Klebsiella pneumoniae* and carbapenem-resistant *Acinetobacter baumannii*. Overall, a KPC-producing *Klebsiella pneumoniae* and/or carbapenem-resistant *Acinetobacter baumannii* bloodstream infection and/or pneumonia occurred in 20% of COVID-19 patients. Superinfections by carbapenem-resistant *Acinetobacter baumannii* in ICU patients with COVID-19 have been reported worldwide [24]. A multicentric Italian study reported an increase in colonization with carbapenem-resistant *Acinetobacter baumannii* from 5.1 per 10,000 ICU-patient-days during January-April 2019 to 26.4 per 10,000 ICU-patient-days during January–April 2020, with a predominance of OXA-23 producers [27]. This massive increase in incidence among ICU patients coincided with the worst phases of the COVID-19 pandemic. Moreover, according to a recent report [33], OXA-23 can be considered the main mechanism involved in carbapenem resistance in *Acinetobacter baumannii* strains isolated from COVID-19 patients in our hospital setting.

Herein, the features most strongly associated with in-hospital mortality were a higher Charlson comorbidity index, lymphopenia at admission, invasive mechanical ventilation/ECMO ≥96 h, and treatment with systemic corticosteroids. The role of comorbidity statuses is in line with previous reports [22]. Likewise, lymphopenia at admission has been previously reported as being significantly associated with the progression to severe disease (OR 4.20, CI 95% 3.46–5.09) and death (OR, 3.71, CI 95% 1.63–8.44) [34]. Lymphopenia is not uncommon among COVID-19 patients and strongly correlates with critical illness. Such a deficiency may contribute to the cytokine storms and higher tissue damage observed in severe COVID-19 infections, as well as to a delay in infection clearance [35]. The use of systemic corticosteroids was found to be a very common and recommended practice, especially during the second COVID-19 wave. Indeed, the use of dexamethasone resulted in lower 28-day mortality, especially in patients receiving invasive mechanical ventilation [36,37]. Nasir et al. showed that systemic steroid use was higher in COVID-19 patients with bacterial co-infections (92% vs. 62% respectively; *p*-value = 0.001) compared to patients without and that steroid use was associated with an increased risk of subsequent infections [38]. However, a study of 226 hospitalized COVID-19 patients showed that although steroid use was associated with an increased incidence of superinfections, the mortality rate was not significantly affected [39]. Our cohort of COVID-19 patients belongs to the first wave when the evidence on COVID-19 patient management was limited. Therefore, the association of systemic corticosteroids with increased mortality may be due to their exclusive use in patients with a severe baseline prognosis.

Previous meta-analyses revealed bacterial co-infections and superinfections to be significant mortality predictors in COVID-19 patients [5,22,23]. The mortality rates reported in COVID-19 patients with bacterial co-infections and/or secondary bacterial infections ranged from 6.5% to 66.7%; however, the observation periods and populations varied, which may explain the wide variation in rates [40]. In addition to increased mortality, other notable trends among COVID-19 patients with bacterial superinfection included a prolonged length of hospital stay, more frequent admission to the ICU, and the use of invasive mechanical ventilation [40].

In our study, the multivariate logistic regression showed no statistically significant contribution of colonization and superinfections by KPC-producing *Klebsiella pneumoniae* and/or carbapenem-resistant *Acinetobacter baumannii* on in-hospital mortality. ~In agreement with our results, Pasero et al. reported that multidrug-resistant infections in COVID-19 ICU patients (mainly Gram-negative bacteria) were associated with longer lengths of stay but not with higher mortality [40]. Likewise, Karrulli et al. showed that 50% of COVID-19 ICU patients developed a multidrug-resistant infection, predominantly bloodstream infections and pneumonia caused by carbapenem-resistant *Klebsiella pneumoniae* and carbapenem-resistant *Acinetobacter baumannii*. However, these were not associated with higher mortality [15]. On the one hand, it is possible that the severity of the SARS-CoV-2 infection overshadowed many other basic characteristics, particularly in the early stages of the pandemic, when the best course of action was still far from clear. Indeed, a prospective study involving 48.902 patients admitted to 260 hospitals in England, Scotland, and Wales in the first wave of the COVID-19 pandemic (admitted from February and June 2020) showed that co-infections or secondary infections were not associated with increased mortality among ICU patients [41]. Assessing the association between a co-infection and an outcome using observational data is complex. First, in the first wave of the pandemic, due to the severity of the SARS-CoV-2 infection, a higher proportion of deaths within the study occurred in the early stages of hospitalization; therefore, these patients had less time to develop superinfections and to obtain microbiological investigation results. In addition, the successful implementation of rapid diagnostics coupled with infectious counselling intervention for the timely and appropriate use of new drugs such as ceftazidime/avibactam and cefiderocol may suffice to counteract the putative positive influence of such infections on mortality [34,42,43].

Our study has some limitations. First, it is a single-center study conducted in a setting with a high diffusion of multidrug-resistant organisms. Second, the sample size is relatively small and the patients were enrolled during only first four months of the SARS-CoV-2 pandemic. Third, the superinfections were sustained by non-multidrug-resistant bacteria, multidrug-resistant species other than carbapenemase-producing Enterobacterales and carbapenem-resistant *Acinetobacter baumannii*, and other pathogens such as viruses and fungi were not included in the analysis, and this represents a potential bias.

## 5. Conclusions

This study assessed the burden of colonization and infections caused by the main multidrug-resistant Gram-negative bacteria in a cohort of patients hospitalized for COVID-19 during the first wave in Italy. A higher comorbidity status, the presence of lymphopenia at admission, undergoing invasive mechanical ventilation/ECMO ≥96 h, and being treated with systemic corticosteroids were the sole features herein observed as significantly associated with in-hospital mortality. Although KPC-producing *Klebsiella pneumoniae* and/or carbapenem-resistant *Acinetobacter baumannii* rectal carriage and/or infections were diagnosed in a remarkable percentage of COVID-19 patients, their impact on in-hospital mortality was not significant. Further studies are needed to assess the burden of antimicrobial resistance as a legacy of COVID-19 in order to identify future prevention opportunities.

## Figures and Tables

**Table 1 viruses-15-01934-t001:** Overall SARS-CoV-2 infection cohort summary statistics.

Patient Characteristics (*n* = 188)	
Age, median [IQR] (years)	69 [67–75]
Male	71%
Community-acquired SARS-CoV-2 infection	65%
Chronic heart disease	66%
Chronic pulmonary disease	22%
Chronic kidney disease	19%
Chronic liver disease	7%
Neoplasia	18%
Solid organ transplant recipient	5%
Diabetes	27%
Obesity	20%
Autoimmune disease	3%
Charlson comorbidity index, median [IQR]	4 [2–5]
Critically ill patient	61%
ICU length of stay, median [IQR] (days)	10 [5–18]
Total hospital length of stay, median [IQR] (days)	29 [18–46]
Clinical presentation	
Unilateral pneumonia	2%
Bilateral pneumonia or ARDS	98%
Pulmonary embolism	10%
D-dimer (ng/mL)	1496 [845–4396]
LDH (UI/L)	644 [458–814.25]
CPK (UI/L)	78 [37–215]
NT-proBNP (pg/mL)	725 [244–3222]
Troponin T (ng/L)	22 [11–53]
Ferritin (ng/mL)	1081.5 [551–1922]
Creatinine (mg/dL)	1.06 [0.74–1.45]
Lymphocytes count (10^9^/L)	0.83 [0.5–1.2]
Procalcitonin (ng/mL)	0.33 [0.1–1.0]
CRP (mg/L)	74.5 [28–139]
Support and management	
Invasive mechanical ventilation or ECMO ≥ 96 h	50%
NIV and/or O_2_ therapy	86%
Systemic corticosteroids	81%
Tocilizumab	8%
Hydroxychloroquine	13%
Lopinavir/ritonavir	11%
Remdesivir	7%
Antimicrobial treatment	96%
Multidrug-resistant bacteria colonization and superinfection	
KPC-Kp and/or CR-ACB rectal carriage	30%
KPC-Kp rectal carriage only	18%
CR-ACB rectal carriage only	3%
KPC-Kp + CR-ACB rectal carriage	10%
KPC-Kp and/or CR-ACB BSI and/or pneumonia	20%
KPC-Kp and/or CR-ACB BSI	11%
KPC-Kp and/or CR-ACB pneumonia	12%
KPC-Kp BSI or pneumonia only	8%
CR-ACB BSI or pneumonia only	5%
KPC-Kp + CR-ACB BSI or pneumonia	6%
Outcome	
In-hospital death	44%

Abbreviations: IQR: interquartile range; ICU: intensive care unit; ARDS: acute respiratory distress syndrome; LDH: lactate dehydrogenase; CPK: creatine phosphokinase; CRP: C-reactive protein; ECMO: Extracorporeal Membrane Oxygenation; NIV: non-invasive ventilation; KPC-Kp: Klebsiella pneumoniae carbapenemase-producing *Klebsiella pneumoniae*; CR-ACB: carbapenem-resistant *Acinetobacter baumannii*; BSI: bloodstream infection.

**Table 2 viruses-15-01934-t002:** Comparison of SARS-CoV-2 survivors vs. non-survivors clinical features.

	Alive|Dead	Mann–Whitney U Test W Statistic|*p*-Value or Fisher’s Exact Test Odds Ratio [95% C.I.]|*p*-Value
**Patient characteristics (*n* = 188)**		
Age, median [IQR] (years)	**67 [55–73]|73 [65–77]**	**W = 2989, *p* < 0.01**
Male	70%|72%	1.09 [0.55–2.18]|*p* = 0.87
Community-acquired SARS-CoV-2 infection	64%|67%	1.18 [0.61–2.67]|*p* = 0.64
Chronic heart disease	60%|72%	1.60 [0.83–315]|*p* = 0.16
Chronic pulmonary disease	**15%|30%**	**2.39 [1.12–5.23]|*p* = 0.02**
Chronic kidney disease	15%|23%	1.65 [0.74–3.71]|*p* = 0.19
Chronic liver disease	6%|10%	1.75 [0.51–6.42]|*p* = 0.40
Neoplasia	**12%|25%**	**2.39 [1.05–5.60]|*p* = 0.03**
Solid organ transplant recipient	6%|8%	0.84 [0.17–3.67]|*p* = 1
Diabetes	23%|31%	1.54 [0.76–3.11]|*p* = 0.24
Obesity	18%|22%	1.25 [0.57–2.74]|*p* = 0.58
Autoimmune disease	4%|2%	0.62 [0.06–4.49]|*p* = 0.69
Charlson comorbidity index, median [IQR]	**3 [2–5]|4 [3–6]**	**W = 3016.5, *p* < 0.01**
Critically ill patient	**53%|71%**	**2.14 [1.12–4.16]|*p* = 0.01**
ICU length of stay, median [IQR] (days)	**2 [0–11]|9 [0–17]**	**W = 3342.5, *p* < 0.01**
Total hospital length of stay, median [IQR] (days)	**33 [23–57]|21 [15–35]**	**W = 5940.5, *p* < 0.01**
**Clinical presentation**		
Unilateral pneumonia	4%|0%	0 [0–1.90]|*p* = 0.13
Bilateral pneumonia or ARDS	**83%|95%**	**4.06 [1.16–17.19]|*p* = 0.01**
Pulmonary embolism	11%|8%	0.72 [0.23–2.09]|*p* = 0.62
D-dimer (ng/mL), median [IQR]	1390 [774–3902]|2066 [877–5141]	W = 3856.5, *p* = 0.17
LDH (UI/L), median [IQR]	639 [480–839]|664 [444–809]	W = 4422, *p* = 0.86
CPK (UI/L), median [IQR]	69 [36–173]|80 [38–282]	W = 3973, *p* = 0.3
NT-proBNP (pg/mL), median [IQR]	**617 [197–1751]|996 [371–5900]**	**W = 3473.5, *p* = 0.01**
Troponin T (ng/L), median [IQR]	20 [9–67]|24 [14–52]	W = 3862, *p* = 0.18
Ferritin (ng/mL), median [IQR]	953 [555–1830]|1150 [537–1953]	W = 3968, *p* = 0.29
Creatinine (mg/dL), median [IQR]	**0.93 [0.72–1.29]|1.16 [0.76–1.53]**	**W = 3540.5, *p* = 0.02**
Lymphocytes count (10^9^/L), median [IQR]	**1 [0.54–1.39]|0.61 [0.44–1.06]**	**W = 5548.5, *p* < 0.01**
Procalcitonin (ng/mL), median [IQR]	0.34 [0.1–1.0]|0.31 [0.15–0.99]	W = 4019.5, *p* = 0.36
CRP (mg/L), median [IQR]	85 [27.7–140]|70.5 [27.8–70.5]	W = 4515.5, *p* = 0.67
**Support and management**		
Mechanical ventilation or ECMO ≥ 96 h	**37%|66%**	**3.30 [1.74–6.36]|*p* < 0.01**
NIV and/or O_2_ therapy	**80%|93%**	**3.19 [1.17–10.18]|*p* = 0.02**
Systemic corticosteroids	**72%|92%**	**4.11 [1.64–11.82]|*p* < 0.01**
Tocilizumab	7%|10%	1.49 [0.45–5.06]|*p* = 0.58
Hydroxychloroquine	10%|14%	1.31 [0.50–3.39]|*p* = 0.66
Lopinavir/ritonavir	8%|14%	2.04 [0.72–6.08]|*p* = 0.15
Remdesivir	9%|6%	0.69 [0.17–2.39]|*p* = 0.58
Antimicrobial treatment	95%|98%	2.02 [0.32–21.71]|*p* = 0.46
**MDR bacteria colonization and superinfection**		
KPC-Kp and/or CR-ACB rectal carriage	28%|34%	1.33 [0.68–2.61]|*p* = 0.42
KPC-Kp rectal carriage only	15%|20%	1.43 [0.63–3.27]|*p* = 0.44
CR-ACB rectal carriage only	4%|1%	0.31 [0.01–3.21]|*p* = 0.38
KPC-Kp + CR-ACB rectal carriage	9%|12%	1.46 [0.50–4.29]|*p* = 0.47
KPC-Kp and/or CR-ACB BSI and/or pneumonia	17%|23%	1.43 [0.65–3.15]|*p* = 0.35
KPC-Kp and/or CR-ACB BSI	9%|13%	1.63 [0.58–4.70]|*p* = 0.34
KPC-Kp and/or CR-ACB pneumonia	8%|17%	2.45 [0.90–7.13]|*p* = 0.06
KPC-Kp BSI or pneumonia only	10%|6%	0.61 [0.16–2.06]|*p* = 0.42
CR-ACB BSI or pneumonia only	5%|5%	1.01 [0.19–4.88]|*p* = 1
KPC-Kp + CR-ACB BSI or pneumonia	3%|10%	3.60 [0.83–21.78]|*p* = 0.06

Bold characters denote statistical significance at *p*-level < 0.05. Abbreviations: IQR: interquartile range; ICU: intensive care unit; ARDS: acute respiratory distress syndrome; LDH: lactate dehydrogenase; CPK: creatine phosphokinase; CRP: C-reactive protein; ECMO: Extracorporeal Membrane Oxygenation; NIV: non-invasive ventilation; MDR: multidrug-resistant; KPC-Kp: KPC-producing *Klebsiella pneumoniae*; CR-ACB: carbapenem-resistant *Acinetobacter baumannii*; BSI: bloodstream infection.

**Table 3 viruses-15-01934-t003:** Logistic regression analyses of in-hospital mortality risk factors.

Feature	Univariate Logistic Regression				Multivariable Logistic Regression			
	*p*-Value	Odds Ratio	95% CI Upper	95% CI Lower	*p*-Value	Odds Ratio	95% CI Upper	95% CI Lower
**Age**	**<0.01**	**1.05**	**1.07**	**1.04**	0.06	1.04	1.06	1.01
Male	0.78	1.09	1.51	0.78				
Community-acquired SARS-CoV-2 infection	0.60	1.17	1.60	0.86				
Chronic heart disease	0.13	1.60	2.20	1.16				
**Chronic pulmonary disease**	**<0.05**	**2.39**	**3.44**	**1.66**	0.44	1.45	2.39	0.88
Chronic kidney disease	0.18	1.65	2.40	1.13				
Chronic liver disease	0.31	1.76	3.08	1.00				
**Neoplasia**	**<0.05**	**2.39**	**3.53**	**1.62**	0.20	2.26	4.31	1.18
Solid organ transplant recipient	0.78	0.83	1.62	0.43				
Diabetes	0.19	1.53	2.14	1.10				
Obesity	0.53	1.25	1.81	0.86				
Autoimmune disease	0.59	0.62	1.50	0.25				
**Charlson comorbidity index**	**<0.05**	**1.27**	**1.36**	**1.18**	**0.01**	**1.41**	**1.59**	**1.24**
**Critically ill patient**	**<0.05**	**2.15**	**2.93**	**1.57**	0.23	0.41	0.87	0.19
**ICU length of stay**	**<0.05**	**1.02**	**1.03**	**1.01**	-	-	-	-
**Total hospital length of stay**	**<0.01**	**0.98**	**0.98**	**0.97**	-	-	-	-
Unilateral pneumonia	0.98	0	-	0				
**Bilateral pneumonia or ARDS**	**<0.05**	**4.08**	**7.25**	**2.30**	0.82	0.84	1.84	0.38
Pulmonary embolism	0.50	0.71	1.17	0.43				
D-dimer	0.40	1.00	1.00	1.00				
LDH	0.55	1.00	1.00	0.99				
CPK	0.06	1.00	1.00	1.00				
NT-proBNP	0.44	1.00	1.00	1.00				
Troponin T	0.21	0.99	0.99	0.99				
Ferritin	0.40	1.00	1.00	0.99				
Creatinine	0.74	1.02	1.11	0.94				
**Lymphocytes count**	**<0.01**	**0.48**	**0.62**	**0.37**	**0.03**	**0.54**	**0.72**	**0.40**
Procalcitonin	0.59	1.00	1.02	0.99				
CRP	0.76	0.99	1.00	0.99				
**Invasive mechanical ventilation or ECMO ≥ 96 h**	**<0.01**	**3.32**	**4.52**	**2.44**	**0.01**	**6.34**	**12.62**	**3.18**
NIV and/or O_2_ therapy	**<0.05**	**3.20**	**5.23**	**1.96**	0.19	2.71	5.83	1.26
Hydroxychloroquine	0.53	1.30	2.02	0.84				
**Systemic glucocorticoids**	**<0.01**	**4.14**	**6.50**	**2.63**	**0.01**	**4.67**	**9.05**	**2.41**
Tocilizumab	0.45	1.49	2.56	0.87				
Lopinavir/ritonavir	0.13	2.04	3.31	1.26				
Remdesivir	0.51	0.68	1.21	0.38				
Antimicrobial treatment	0.40	2.02	4.73	0.86				
KPC-Kp and/or CR-ACB rectal carriage	0.36	1.33	1.83	0.97				
KPC-Kp rectal carriage only	0.34	1.43	2.10	0.97				
CR-ACB rectal carriage only	0.29	0.30	0.95	0.09				
KPC-Kp + CR-ACB rectal carriage	0.43	1.46	2.37	0.89				
KPC-Kp and/or CR-ACB BSI and/or pneumonia	0.32	1.43	2.07	0.99				
KPC-Kp and/or CR-ACB BSI	0.30	1.62	2.62	1.01				
KPC-Kp and/or CR-ACB pneumonia	0.05	2.46	3.93	1.53				
KPC-Kp BSI or pneumonia only	0.38	0.60	1.07	0.34				
CR-ACB BSI or pneumonia only	0.98	1.01	2.01	0.50				
KPC-Kp + CR-ACB BSI or pneumonia	0.06	3.62	7.25	1.81				

Bold characters denote statistical significance at *p*-level < 0.05. Abbreviations: CI: confidence interval; ICU: intensive care unit; ARDS: acute respiratory distress syndrome; LDH: lactate dehydrogenase; CPK: creatine phosphokinase; CRP: C-reactive protein; ECMO: Extracorporeal Membrane Oxygenation; NIV: non-invasive ventilation; KPC-Kp: KPC-producing *Klebsiella pneumoniae*; CR-ACB: carbapenem-resistant *Acinetobacter baumannii*; BSI: bloodstream infection.

## Data Availability

The authors confirm that the data supporting the findings of this study are available within the article.

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
