# Peer review of "Prevalence and Impact on Mortality of Colonization and Super-Infection by Carbapenem-Resistant Gram-Negative Organisms in COVID-19 Hospitalized Patients"

_viruses, 2023, doi:10.3390/v15091934_

Round 1
Reviewer 1 Report (Previous Reviewer 3)
The manuscript was improved according to the reviewers' recommendation, I agree with this version to be published
Author Response
We thank the reviewer for these accurate appraisals.
Reviewer 2 Report (Previous Reviewer 2)
I am satisfied with modifications.
Author Response
We thank the reviewer for this comment.
Reviewer 3 Report (Previous Reviewer 1)
This study focuses on the presence and impact of carbapenem-resistant pathogens in COVID-19 patients, specifically KPC-producing Klebsiella pneumoniae and carbapenem-resistant Acinetobacter baumannii. Utilizing a retrospective observational design, the study indeed finds a notable prevalence of rectal carriage of these pathogens in COVID-19 patients, accounting for 30%. Surprisingly, however, these pathogens do not show a significant association with in-hospital mortality. Additionally, the study reveals multiple factors primarily influencing in-hospital mortality, including a high Charlson comorbidity index, lymphopenia, invasive mechanical ventilation or ECMO for more than 96 hours, and systemic corticosteroid treatment.
The study offers rich information in multiple aspects, especially regarding the prevention of antimicrobial resistance. While numerous related studies have been published, this study does not present any novel findings. Moreover, the study is conducted in a single medical center with a relatively small sample size, which may limit the generalizability of the results. It is advisable to compare the findings more extensively with the existing literature to enhance the reliability of the research outcomes.
Author Response
We thank the reviewer for these accurate appraisals. All limitations listed by the reviewer have been made explicit in the text. The discussion was expanded and the results compared with the existing literature.
This manuscript is a resubmission of an earlier submission. The following is a list of the peer review reports and author responses from that submission.
Round 1
Reviewer 1 Report
This study deeply explores the colonization and infection of carbapenem-resistant Gram-negative bacteria such as CPE and CR-ACB during the first wave of COVID-19 in Italy. It outlines the characteristics of 188 patients with a median age of 69. Thirty percent of COVID-19 patients were rectal carriers of KPC-Kp and/or CR-ACB. Twenty percent of colonized patients later developed pneumonia and/or bloodstream infections, and the study did not find a significant statistical correlation between their colonization or superinfection and in-hospital mortality rates. This research emphasizes the mortality risk factors in COVID-19 hospitalized patients and reveals that colonization or superinfection by carbapenem-resistant Gram-negative bacteria may not be the main cause of death in these patients. As the author points out, there is already extensive literature exploring this subject. Additionally, as a single-center study, its limitations include constraints on sample size, which may affect the generalizability and reliability of the results. There was no mention of the control background values in this hospital, nor the proportion of hospitalized patients with KPC-Kp and/or CR-ACB in this hospital. The novelty of this study still needs to be strengthened.
Reviewer 2 Report
Dear authors,
I believe some major changes must be done in order for the manuscript to be appropriate for Viruses Journal.
There are abbreviations mentioned into the text and not explain, such as in abstract, line 33. The explanation is in line 163.
There for I believe a list of abbreviations will be suitable in the beginning of the manuscript as a table.
The introduction is too short, more information are needed, regarding the gut microbiome, multi-drug-resistant bacteria, also COVID19 implications in both.
In methods
Study design would benefit of a chart flow. Microbiological diagnosis may be extended.
Result section can be improved, MDR may be detailed.
Overall some improvement must be made. The manuscript is interesting and touches a hot subject, such as MDR.
Reviewer 3 Report
In general, the manuscript is well prepared, with interesting results regarding co-infections of SARS CoV2 and multiresistant bacteria. In order to improve the quality of the manuscript the authors should write all the bacteria names properly, in italics. In addition, If is possible improve the table presentations too, for a more clear reading to the possible readers.